# Prioritized SNP Selection from Whole-Genome Sequencing Improves Genomic Prediction Accuracy in Sturgeons Using Linear and Machine Learning Models

**DOI:** 10.3390/ijms26147007

**Published:** 2025-07-21

**Authors:** Hailiang Song, Wei Wang, Tian Dong, Xiaoyu Yan, Chenfan Geng, Song Bai, Hongxia Hu

**Affiliations:** 1Fisheries Science Institute, Beijing Academy of Agriculture and Forestry Sciences & Beijing Key Laboratory of Fisheries Biotechnology, Beijing 100068, China; songhailiang@baafs.net.cn (H.S.);; 2Key Laboratory of Sturgeon Genetics and Breeding, Ministry of Agriculture and Rural Affairs, Hangzhou 311799, China

**Keywords:** sturgeon, genomic prediction, GWAS, machine learning models, SNP density, aquaculture industry, breeding, caviar traits

## Abstract

Genomic prediction has emerged as a powerful tool in aquaculture breeding, but its effectiveness depends on the careful selection of informative single nucleotide polymorphisms (SNPs) and the application of appropriate prediction models. This study aimed to enhance genomic prediction accuracy in Russian sturgeon (*Acipenser gueldenstaedtii*) by optimizing SNP selection strategies and exploring the performance of linear and machine learning models. Three economically important traits—caviar yield, caviar color, and body weight—were selected due to their direct relevance to breeding goals and market value. Whole-genome sequencing (WGS) data were obtained from 971 individuals with an average sequencing depth of 13.52×. To reduce marker density and eliminate redundancy, three SNP selection strategies were applied: (1) genome-wide association study (GWAS)-based prioritization to select trait-associated SNPs; (2) linkage disequilibrium (LD) pruning to retain independent markers; and (3) random sampling as a control. Genomic prediction was conducted using both linear (e.g., GBLUP) and machine learning models (e.g., random forest) across varying SNP densities (1 K to 50 K). Results showed that GWAS-based SNP selection consistently outperformed other strategies, especially at moderate densities (≥10 K), improving prediction accuracy by up to 3.4% compared to the full WGS dataset. LD-based selection at higher densities (30 K and 50 K) achieved comparable performance to full WGS. Notably, machine learning models, particularly random forest, exceeded the performance of linear models, yielding an additional 2.0% increase in accuracy when combined with GWAS-selected SNPs. In conclusion, integrating WGS data with GWAS-informed SNP selection and advanced machine learning models offers a promising framework for improving genomic prediction in sturgeon and holds promise for broader applications in aquaculture breeding programs.

## 1. Introduction

Sturgeons (Acipenseridae) represent one of the most ancient lineages of vertebrates, having persisted for over 200 million years [1]. Today, they hold exceptional economic value, primarily for their roe, which is processed into caviar. Caviar is often referred to as “black gold” due to its limited availability, labor-intensive production process, and consistently high market demand. Among caviar products, golden eggs are particularly prized, commanding premium prices in international markets [2]. In addition to roe, sturgeon meat is highly regarded for its rich nutritional value, further driving demand for sturgeon aquaculture [2]. As a result, caviar yield, caviar color, and body weight are considered key economic traits in sturgeon breeding [3]. China has become the global leader in sturgeon aquaculture and caviar export, accounting for 85% of global sturgeon production and 50% of the global caviar market in 2024 [4].

However, conventional breeding in sturgeon is hindered by long generation intervals, typically requiring 6–8 years to reach sexual maturity [3]. This slow reproductive cycle limits genetic gain and highlights the need for genomic-based approaches to accelerate breeding progress. Genomic selection (GS) [5], which uses genome-wide SNPs to predict genomic estimated breeding values (GEBVs), has proven effective in other aquaculture species [6,7]. It improves accuracy over pedigree-based selection by capturing linkage disequilibrium (LD) between markers and quantitative trait loci (QTLs) [5]. While linear models such as genomic best linear unbiased prediction (GBLUP) are commonly used [8], machine learning methods have recently gained attention for their potential to model complex genetic architectures beyond linear assumptions [9].

The rapid advancement of sequencing technologies and the continuous reduction in costs have made whole-genome sequencing (WGS) increasingly accessible for a wide range of aquaculture species. In theory, WGS provides exhaustive genome-wide coverage by capturing millions of SNPs, thereby offering the potential to tag all QTLs and improve the accuracy of genomic predictions beyond that achieved with conventional SNP arrays [10,11]. However, numerous studies have shown that the inclusion of all WGS-derived variants does not necessarily enhance genomic prediction accuracy [12,13]. On the contrary, the presence of extensive redundant markers often introduces noise, mainly caused by non-informative or highly correlated variants, which can obscure true genetic signals and potentially reduce the accuracy of genomic prediction [14,15]. In addition, the sheer volume of WGS data—often comprising tens of millions of SNPs—creates substantial computational challenges, particularly for Bayesian and machine learning-based models, which struggle to process such high-dimensional datasets effectively. These issues highlight the importance of developing robust SNP selection strategies that enrich for informative variants while minimizing redundancy. An optimized marker panel not only alleviates computational burdens but also enables the effective application of advanced predictive models, thereby maximizing the benefits of WGS in genomic selection programs.

This study aims to enhance genomic prediction accuracy for three key traits in Russian sturgeon, namely caviar yield, caviar color, and body weight, by optimizing SNP selection and prediction models. We systematically evaluated GWAS-based prioritization, LD pruning, and random sampling to reduce marker redundancy, and assessed the performance of linear and machine learning models across varying SNP densities to identify the most effective method. This work is novel in its integrated use of SNP selection and machine learning methods in sturgeon, a species with long generation intervals, providing a practical strategy to accelerate breeding progress.

## 2. Results

### 2.1. Summary Statistics of Key Phenotypic Traits

Descriptive statistics for caviar yield, caviar color, and body weight in 971 individuals are presented in Table 1. Caviar yield, calculated as the ratio of total caviar weight to body weight, showed a mean of 0.193 with a standard deviation (SD) of 0.057 and a coefficient of variation (CV) of 29.41%. Caviar color was assessed on a subjective four-point scale, with scores ranging from 1 (dark) to 4 (gold). The average color score was 2.398, with an SD of 0.642 and a CV of 26.79%, suggesting substantial variation in pigmentation. Body weight had a mean of 19.806 kg, ranging from 9.700 to 116.800 kg, with an SD of 5.096 and a CV of 25.73%, reflecting high variability in growth traits within the population.

### 2.2. Whole-Genome Sequencing, Population Structure and Linkage Disequilibrium

Whole-genome sequencing of 971 fish generated a total of 94.93 billion reads, with an average of 0.10 billion reads per individual. A high mapping rate of 92.44% was achieved, resulting in an average sequencing depth of 13.52× (range: 5.06× to 25.85×). After stringent quality control, 8.22 million high-confidence SNPs were retained for subsequent analyses. SNP density varied markedly across chromosomes, from 317 (Chr60) to 655,341 (Chr3), with an average of 4171.98 SNPs per megabase (Figure 1A).

Principal component analysis (PCA) based on genome-wide SNP data revealed that the majority of individuals clustered closely, while a subset showed clear separation along the top three principal components, suggesting the presence of population stratification (Figure 1B). This highlights the need to include principal components as covariates in genome-wide association models to correct for potential confounding. Analysis of linkage disequilibrium (LD) indicated extensive marker redundancy, with 4,978,819 SNP pairs exhibiting r^2^ > 0.2 (Figure 1C).

### 2.3. Impact of SNP Selection Strategies and SNP Density on Genomic Prediction

We assessed the influence of SNP selection strategies—GWAS-based selection, LD pruning, and random sampling—across varying marker densities for caviar yield, caviar color, and body weight (Figure 2, Figure 3 and Figure 4). At SNP densities below 10 K, all three strategies produced lower genomic prediction accuracies compared to the WGS-based GBLUP model. For example, for caviar yield, prediction accuracies were 0.304 (GWAS-based), 0.329 (LD pruning), and 0.341 (random sampling), all below the WGS-based GBLUP accuracy of 0.368 (Figure 2). Similar trends were observed for caviar color and body weight (Figure 3 and Figure 4), highlighting the advantage of comprehensive genome coverage at low marker densities.

In contrast, when SNP density reached or exceeded 10 K, GWAS-based selection consistently outperformed WGS-based GBLUP across all traits. For instance, the prediction accuracy for caviar yield increased to 0.391 with GWAS-based selection, surpassing that of WGS-based GBLUP, whereas LD pruning (0.353) and random sampling (0.352) remained slightly lower (Figure 2). A similar trend was observed for caviar color, with GWAS-based selection achieving an average 3.3% improvement in accuracy over WGS-based GBLUP at higher densities (Figure 3). For body weight, GWAS-selected SNP panels also yielded higher accuracies than WGS data when SNP density was ≥10 K (Figure 4). Notably, when SNP density exceeded 10 K—particularly at 30 K and 50 K—LD-pruned and even randomly sampled SNPs achieved prediction accuracies comparable to those of the full WGS model, suggesting a high degree of redundancy among WGS markers.

Regarding prediction bias, mean squared error (MSE), and mean absolute error (MAE), most SNP selection strategies produced results comparable to those obtained with WGS-based GBLUP across all densities and traits (Appendix A). A notable exception occurred at very low SNP densities (1 K and 5 K), where the GWAS-based strategy exhibited inflated prediction bias (Appendix A), suggesting a tendency to overestimate genetic values due to insufficient marker information. As SNP density increased, prediction bias, MSE, and MAE stabilized across all methods, demonstrating the robustness of GBLUP to different SNP selection strategies at higher marker densities.

The proportion of phenotypic variance explained (PVE) by SNPs selected using different strategies across increasing marker densities is summarized in Table 2. Overall, the GWAS-based strategy consistently outperformed LD-based pruning and random sampling across all SNP densities and traits. For all three traits—caviar yield, caviar color, and body weight—the PVE increased substantially with higher SNP densities under the GWAS strategy, reaching 5.25%, 4.76%, and 6.31%, respectively at 50 K SNPs. In contrast, LD-based and random strategies explained substantially less variance, particularly at low SNP densities, where PVE values remained below 0.1% in most cases.

### 2.4. Impact of Linear and Machine Learning Models on Genomic Prediction

Using GWAS-selected SNP sets with densities ≥ 10 K, we assessed the performance of linear and machine learning models across caviar yield, caviar color, and body weight (Figure 5). Among linear models, GBLUP, BayesA, BayesB, and BayesLasso showed similar prediction accuracies, while BayesCπ consistently performed slightly worse (e.g., 0.354 vs. 0.391 for caviar yield using 10 K SNPs). Among machine learning methods, RF consistently achieved the highest accuracy across all traits, outperforming both linear and other machine learning models by 2.3%, 1.2%, and 0% for caviar yield, color, and body weight, respectively. Compared to other machine learning models—including SVR, KRR, and XGB—RF further improved prediction accuracy by 2.9%, 2.2%, and 1.2%, respectively (Figure 5).

In addition to the GWAS-based marker sets, we further examined model performance under LD-pruned and randomly selected SNP subsets with densities ≥ 10 K (Figure 5). For caviar yield, KRR outperformed other machine learning models under the LD-pruned strategy. Conversely, under the random selection strategy, all models—including both linear and machine learning models—exhibited prediction accuracies that were equal to or lower than those obtained by WGS-GBLUP. For caviar color, no model—regardless of algorithm or selection strategy—surpassed the performance of WGS-GBLUP under either LD or random selection. Interestingly, for body weight, support vector regression (SVR) achieved the highest prediction accuracy across all models and SNP selection strategies. Notably, SVR consistently outperformed WGS-GBLUP even at low marker densities (1 K and 5 K), highlighting its robustness and capacity to capture complex trait architecture under sparse genomic information. Notably, across all scenarios, Bayesian and machine learning models consistently yielded lower mean squared error (MSE) and mean absolute error (MAE) than GBLUP, regardless of SNP selection strategy or marker density (Appendix A). This indicates that these models provide a superior fit to the data.

## 3. Discussion

Previous studies have indicated that directly utilizing all available markers from WGS data does not necessarily improve, and may even reduce, the accuracy of genomic prediction, due to the inclusion of redundant, low-frequency, or non-informative variants. For example, Perez-Enciso et al. [12] demonstrated through simulation that adding all sequence variants could deteriorate prediction accuracy by introducing noise. In Japanese flounder, Lu et al. [13] showed that pre-selecting informative SNPs outperformed models using the full WGS dataset for disease-resistance traits. Similarly, Zhang et al. [14] reported that using all WGS variants did not consistently outperform lower-density SNP chips for predicting feed efficiency in pigs. Building on this understanding, we implemented three distinct SNP selection strategies—GWAS-based prioritization, linkage disequilibrium (LD) pruning, and random sampling—to remove uninformative markers and reduce redundancy. Among them, GWAS-based selection consistently achieved higher predictive accuracies at moderate to high SNP densities (≥10 K) across all three traits examined. This may be attributed to the greater SNP-trait associations (smaller *p*-values) selected by the GWAS-based selection strategy (Figure 6). This trend aligns with findings in other aquaculture species such as rainbow trout [16] and Asian seabass [17], where the prioritization of trait-associated SNPs has been shown to capture a greater proportion of phenotypic variance, thereby facilitating more accurate genomic predictions (Table 2). However, a critical challenge in GWAS-guided genomic selection lies in determining the appropriate number of SNPs to include. Our previous work in four aquaculture species demonstrated that using only genome-wide significant SNPs (selected by false discovery rate thresholds) combined with GFBLUP models did not consistently enhance prediction accuracy [18]. Similar findings were reported by Vu et al. [19], suggesting that a limited number of highly significant SNPs may not capture sufficient genetic variance. In the present study, by systematically varying SNP density from 1 K to 50 K, we observed that low-density panels failed to maintain predictive power, whereas higher-density panels (≥10 K) significantly improved performance. These results highlight the importance of selecting a sufficiently large set of SNPs when applying GWAS-based strategies to maximize the benefits for genomic prediction.

For the LD pruning strategy, our results demonstrated that when SNP density reached between 30 K and 50 K, the predictive accuracy was comparable to that achieved using the full whole-genome sequencing (WGS) dataset. This finding is consistent with our previous studies [20], suggesting that a moderate number of SNPs is sufficient to maintain high prediction accuracy in sturgeon. LD pruning effectively reduces marker redundancy in WGS data, as genomic selection assumes SNPs are in linkage disequilibrium with causative QTLs. Thus, as long as key QTL-linked variance is captured, moderate-density panels can achieve high accuracy with reduced computational demands. Notably, in this study, the random sampling strategy also achieved prediction accuracies similar to those obtained with WGS when SNP densities ranged between 30 K and 50 K. This may be partially attributed to the fact that, at sufficiently high densities, randomly selected SNPs can collectively explain most of the genetic variance. Another contributing factor might be the limited number of replicates (20 repetitions of fivefold cross-validation) used in this study, which may not have been sufficient to fully capture the variability introduced by random marker selection.

In this study, we systematically evaluated the impact of SNP density (1 K–50 K) on genomic prediction accuracy. Consistent with our previous findings in sturgeon [20], we observed that approximately 50 K SNPs are sufficient to achieve high predictive performance. These results suggest that the use of extremely high-density marker panels is unnecessary for maintaining prediction accuracy. Instead, a moderate density of SNPs appears adequate to capture the majority of genetic variation influencing complex traits. This has important practical implications for breeding programs, as reducing marker density can significantly lower genotyping costs without compromising the efficiency of genomic selection in sturgeon.

We also compared the performance of different linear and machine learning models for genomic prediction. The extremely high marker density generated by whole-genome sequencing data has historically posed challenges for the application of Bayesian and machine learning methods due to computational limitations. By reducing SNP density through targeted selection strategies, we were able to address these challenges and enable direct comparisons among models.

Bayesian models, which allow for heterogeneous distributions of marker effects, are theoretically expected to outperform GBLUP [21,22]. However, our results demonstrated that under the GWAS-based SNP selection strategy with densities exceeding 10 K, Bayesian models yielded prediction accuracies comparable to or slightly lower than those of GBLUP. Consistent patterns were observed across other SNP selection methods and marker densities. A likely explanation is that the traits analyzed here are classical quantitative traits (Appendix A), influenced by a large number of loci each with small effects. Under such genetic architectures, the potential advantages of Bayesian approaches may be minimized, leading to similar predictive performance across models. These findings align with previous studies reporting little to no improvement in genomic prediction accuracy when using Bayesian models over GBLUP for highly polygenic traits [23,24].

In contrast, machine learning methods outperformed linear and Bayesian models in this study, particularly under the GWAS-based SNP selection strategy. These models, capable of capturing complex nonlinear relationships, have shown promise in genomic prediction across aquaculture species. For example, Luo et al. [25] reported that the NeuralNet method showed higher genomic prediction accuracy than GBLUP and BayesB for shrimp growth traits. Nguyen and Vu [26] reported that machine learning methods obtained higher prediction accuracies compared to linear models and Bayesian models for skin fluke disease resistance in yellowtail kingfish. A key factor for achieving high prediction accuracy with machine learning models is the proper tuning of hyperparameters. Our previous studies have shown that hyperparameter optimization significantly improves the performance of machine learning models, as compared to models without tuning [27]. In this study, we employed grid search for hyperparameter optimization to ensure optimal performance.

Moreover, the choice of SNP selection strategy influenced the performance of machine learning models, with the best-performing model varying across strategies. Under the GWAS-based strategy, RF yielded the highest accuracy, while SVR performed better under the LD pruning strategy. These results highlight the distinctive strengths of different machine learning algorithms. RF is particularly adept at handling high-dimensional data and capturing intricate interactions among markers [28], while SVR excels in modeling smaller datasets with fewer features [29]. Consequently, our findings suggest that in practical applications, it is essential to evaluate multiple machine learning models to select the most suitable one for specific genomic prediction tasks. It is important to note that, across all scenarios, both Bayesian and machine learning models demonstrated superior performance in terms of MSE and MAE compared to GBLUP, reflecting their advantages in model fitting (Appendix A). Looking forward, integrating multi-omics data with machine learning models, based on GWAS-driven SNP selection strategies, presents a promising avenue for further improving genomic prediction accuracy.

## 4. Materials and Methods

### 4.1. Sample Collection and Trait Measurement in Russian Sturgeon

The Russian sturgeons utilized in this research were sourced from Hangzhou Qiandaohu Xunlong Sci-tech Co., Ltd. (Hangzhou, China), a representative facility with standardized breeding, controlled environment, and well-documented pedigrees. In 2012, a total of 251 sturgeons (78 females and 173 males) underwent artificial insemination, resulting in the establishment of 192 full-sibling families. The fish were raised under controlled aquaculture conditions, with consistent water quality, temperature, and feeding protocols maintained across tanks. At the typical age of sexual maturity, roe development was evaluated using in vitro puncture. Individuals with an average roe diameter exceeding 2.8 mm were individually tagged using passive integrated transponder markers, and fin tissue samples were preserved in absolute ethanol. These tagged individuals were subsequently processed for caviar production at Hangzhou Qiandaohu Xunlong Sci-tech Co., Ltd. Data on body weight, total caviar weight, and caviar color were recorded for each fish. Caviar yield was computed as the ratio of total caviar weight to body weight, while caviar color was subjectively assessed on a 1 to 4 scale reflecting color depth: gold (4), light (3), medium (2), and dark (1). A single operator consistently assigned all color scores, utilizing a standard reference image for classification. A total of 971 fish with documented phenotypes were selected for subsequent analysis. Descriptive statistics summarizing these phenotypes are presented in Table 1.

### 4.2. Whole-Genome Sequencing, SNP Detection and Quality Control

Genomic DNA was extracted using the phenol–chloroform method. Whole-genome sequencing of 971 individuals was performed on the DNBSEQ-T7 platform (MGI Tech Co., Ltd., Shenzhen, China) with 150 bp paired-end libraries. Reads with Phred scores > 20 were retained and aligned to the sterlet reference genome (*Huso ruthenus*, assembly ASM1064508v1) [30] using the Burrows–Wheeler Aligner (BWA v0.7.17) [31]. Owing to the segmental rediploidization of sturgeon genomes [30] and the absence of a high-quality polyploid reference for Russian sturgeon, the diploid sterlet genome served as a proxy reference in this study. BAM files were generated from SAM files with SAMtools v1.2 [32], and PCR duplicates were marked and removed with Picard tools (http://broadinstitute.github.io/picard/, accessed on 26 October 2023). SNP detection was conducted using the UnifiedGenotyper module in the Genome Analysis Toolkit (GATK v3.5) [33]. To ensure high-quality variant calls, stringent filtering criteria were applied. SNPs were retained if they satisfied the following conditions: FisherStrand < 60, Quality ≥ 50, Quality by Depth ≥ 2.0, and Read Position Rank Sum < −8.0. To increase the power of genomic prediction and address missing data, genotype imputation was conducted using Beagle v5.1 [34] under default settings. Following imputation, quality control (QC) was carried out with PLINK v1.9 [35]. Variants were excluded if they had a call rate below 90%, a minor allele frequency (MAF) less than 0.05, or deviated significantly from Hardy–Weinberg equilibrium (*p* < 1 × 10^−7^). After QC filtering, a total of 8,218,317 high-confidence SNPs were retained for subsequent genomic analyses.

### 4.3. Population Structure and Linkage Disequilibrium Analysis

Principal component analysis (PCA) was conducted to investigate the population stratification among Russian sturgeon individuals based on high-confidence SNP genotypes. The genetic relationship matrix was computed using PLINK v1.9 [35], and the top principal components were extracted via the --pca command. The first three components were plotted to visualize genetic variation and to detect potential population structure or outliers within the dataset. Linkage disequilibrium (LD) analysis was performed using PLINK v1.9 [35] to examine the non-random association between SNPs across the genome. Pairwise LD statistics (r^2^ values) were calculated using the --r2 option with default parameters, considering all SNP pairs within a sliding window. The distribution of LD values provides a reference for marker informativeness in downstream genomic analyses.

### 4.4. Genome-Wide Association Study (GWAS)

GWAS was conducted using a linear mixed model that incorporates both fixed and random effects to evaluate the association of each SNP with traits individually, expressed as [36]y=1μ+Pα+Zg+xb+e,
where y represents the vector of phenotypic values, μ is the overall mean, α is the matrix of the top 10 principal components derived from genome-wide SNP data, included as fixed effects to control for population structure, and P is an incidence matrix linking α to y. b denotes the SNP effect. x is the genotype vector for the SNP being tested (coded as 0, 1, or 2). g is the random vector representing polygenic background effects, assumed to follow a normal distribution N(0, $G\sigma_{a}^{2}$), where G is the genomic relationship matrix computed using all SNP markers based on the method proposed by VanRaden [8], and $\sigma_{a}^{2}$ is the additive genetic variance. Z is the incidence matrix linking phenotypes to polygenic effects, and e is the residual vector, assumed to follow a normal distribution N(0, $I\sigma_{e}^{2}$), where $\sigma_{e}^{2}$ is the residual variance. GWAS was performed using the GEMMA (v0.98) software [37].

For each SNP significantly associated with the trait, the proportion of phenotypic variance explained (PVE) was calculated to quantify the contribution of the SNP to the total phenotypic variation. The PVE for a single SNP was computed using the following formula [38]:$$PVE=\frac{2{\hat{\beta }}^{2}\times MAF\times (1-MAF)}{2{\hat{\beta }}^{2}\times MAF\times \left(1-MAF\right)+{\left[se\left(\hat{\beta }\right)\right]}^{2}\times 2N\times MAF\times (1-MAF)}$$
where $\hat{\beta }$ is the estimated additive effect of the SNP on the trait from the GWAS model, $se(\hat{\beta })$ is the standard error of the estimated effect, MAF is the minor allele frequency of the SNP, and N is the number of individuals included in the GWAS. This formulation corrects for sampling variance and allows a more accurate quantification of SNP contributions to trait variability.

### 4.5. SNP Selection Strategies for Genomic Prediction

Given the high density and redundancy of SNP markers obtained from whole-genome sequencing data, the presence of excessive non-informative markers may negatively impact the accuracy of genomic prediction. To address this issue, we implemented three different SNP selection strategies to evaluate their effects on prediction performance: (1) GWAS-based prioritization, where SNPs significantly associated with the trait were selected based on *p*-values from GWAS, aiming to retain markers most likely to affect the trait; (2) LD-based pruning, where SNPs in high LD were filtered to reduce redundancy and preserve representative, independent markers; and (3) random sampling, where SNPs were randomly selected across the genome without considering trait association or LD, serving as a control strategy.

For each strategy, SNP panels of varying densities (1 K, 3 K, 5 K, 10 K, 30 K, and 50 K) were constructed. The upper limit of 50 K was selected based on our previous findings, which indicated that a density of 50 K SNPs is sufficient to achieve high prediction accuracy in this population [20]. These SNP sets were then used to compare the impact of marker selection strategy and density on the accuracy of genomic prediction.

### 4.6. Genomic Prediction Models

To estimate GEBVs, we employed a set of linear and machine learning-based prediction models, each representing different assumptions about the genetic architecture of the traits.

#### 4.6.1. Linear Models

The linear models included the GBLUP model, which assumes that all markers contribute equally to the genetic variance and that effects are normally distributed, and Bayesian alphabet models (BayesA, BayesB, BayesCπ and BayesLasso), which allow for marker-specific variances and accommodate the possibility that only a subset of markers has large effects.

##### GBLUP

The GBLUP model [8] was used to estimate genomic estimated breeding values (GEBVs) for all genotyped individuals. The model is formulated as follows:y=1μ+Zg+e,
where all variables in this model, including y,μ,Z,g,e are consistent with those used in the GWAS model. The genomic relationship matrix **G** was constructed using SNP markers from whole-genome sequencing data, as well as from selected SNP subsets with varying densities (1 K, 3 K, 5 K, 10 K, 30 K, and 50 K) derived through different marker selection strategies. GBLUP was performed using the DMUAI procedure in DMU v6 software [39].

##### Bayesian Models

The statistical models can be expressed as$$y=1\mu +\sum_{i=1}^{m}Z_{i}g_{i}+e$$
where y, μ, and e are consistent with the GBLUP model. $Z_{i}$ is the i-th SNP genotype vector (e.g., 0, 1, 2); and $g_{i}$ is the effect value of the i-th SNP. GEBVs were calculated as $g={{(Z}^{\prime }Z+I\sigma_{e}^{2}/\sigma_{g_{i}}^{2})}^{-1}Z^{\prime }(y-1\mu )$**.**

Among them, $\sigma_{g_{i}}^{2}$ is the variance of the i-th SNP effect, which is directly related to the genetic structure of traits. The emphasis and difficulty of Bayesian-based models lie in making reasonable assumptions about the prior distribution of hyperparameters, specifically the distribution of $g_{i}$ and its variance. The following are the differences of prior assumptions of several classical Bayesian methods [21,22]:$$g_{i}|\left(\pi ,\sigma_{g}^{2}\right)\sim \left\{\begin{array}{l} 0;\sigma_{g}^{2}=0;\pi \sim {dist}_{0}\\ g_{i}|\sigma_{g}^{2}\sim {dist}_{1};\sigma_{g}^{2}\sim {dist}_{2};(1-\pi ) \end{array}\right.$$$$\sim \left[\begin{array}{l} BayesA:{dist}_{0}=0\;\&\;{dist}_{1}=N(0,\sigma_{g}^{2})\;\&\;{dist}_{2}=\chi^{-2}(\nu ,S)\\ \begin{array}{l} BayesB:{dist}_{0}=0.95\;\&\;{dist}_{1}=N(0,\sigma_{g}^{2})\;\&\;{dist}_{2}=\chi^{-2}(\nu ,S)\\ BayesC\pi :{dist}_{0}=U(0,1)\;\&\;{dist}_{1}=N(0,\sigma_{g}^{2})\;\&\;{dist}_{2}=C\\ BayesLASSO:{dist}_{0}=0\;\&\;{dist}_{1}=N(0,\sigma_{g}^{2})\;\&\;{dist}_{2}=Expon(\lambda^{2}/2) \end{array} \end{array}\right.$$
where ${dist}_{0}$ represents the distribution of non-effect markers; ${dist}_{1}$ denotes the distribution of marker effect values; and ${dist}_{2}$ represents the distribution of marker effect variance. The degree of freedom ν and the scale parameter S are directly related to the genetic structure of traits. For Bayesian models, the Monte Carlo Markov chains (MCMC) were run for 18,000 cycles of Gibbs sampling, with the first 3000 cycles discarded as burn-in. Bayesian models were executed using the BGLR R package (v1.1.4) [40].

#### 4.6.2. Machine Learning Models

In addition to linear models, we evaluated machine learning models to capture potential non-linear relationships between SNP markers and phenotypes. The evaluated models included support vector regression (SVR) [29], random forest (RF) [28], kernel ridge regression (KRR) [41], and extreme gradient boosting (XGB) [42]. Each of these methods leverages distinct modeling strategies—such as kernel-based learning (SVR, KRR) and ensemble learning (RF, XGB)—to capture complex patterns in high-dimensional genomic data. Hyperparameters for each model were optimized through comprehensive grid search combined with repeated fivefold cross-validation within training sets to prevent overfitting and ensure robust performance. All machine learning models were implemented using the Scikit-learn package for Python (v 3.8.3) [43].

It is important to note that the extremely high dimensionality of WGS data, characterized by a vast number of SNP markers, poses substantial computational challenges for Bayesian and machine learning models, rendering them impractical for direct genomic prediction using the full dataset. Therefore, for these models, GEBVs were predicted using SNP subsets selected through three alternative strategies at varying marker densities. The prediction performance of each model and selection strategy was subsequently compared to that of the GBLUP model, which was trained using the full set of WGS-derived SNPs.

### 4.7. Evaluation of Genomic Prediction Accuracy

To assess the predictive performance of genomic models, a fivefold cross-validation scheme was implemented. In each cross-validation iteration, the genotyped population was randomly divided into five approximately equal subsets. Four subsets were used for model training, and the remaining subset served as the validation set. This process was repeated so that each subset was used as the validation set once, and the entire fivefold cross-validation was repeated 20 times to ensure stable and robust evaluation across all scenarios.

Prediction accuracy was quantified by calculating the Pearson correlation coefficient between observed phenotypes and predicted genomic estimated breeding values (GEBVs) in the validation set. In addition, regression of phenotypes on GEBVs was used to assess prediction bias. Model performance was further evaluated using mean squared error (MSE) and mean absolute error (MAE), providing complementary metrics of prediction precision and robustness.

## 5. Conclusions

In conclusion, our study demonstrates that integrating WGS data with appropriate SNP selection strategies and advanced predictive models can significantly enhance genomic prediction accuracy in sturgeon breeding. In particular, GWAS-based SNP selection at moderate densities (≥10 K), combined with machine learning models such as random forest, provided superior performance over traditional methods. These results offer practical guidance for implementing cost-effective and accurate genomic selection in aquaculture breeding programs.

## Figures and Tables

**Figure 1 ijms-26-07007-f001:**
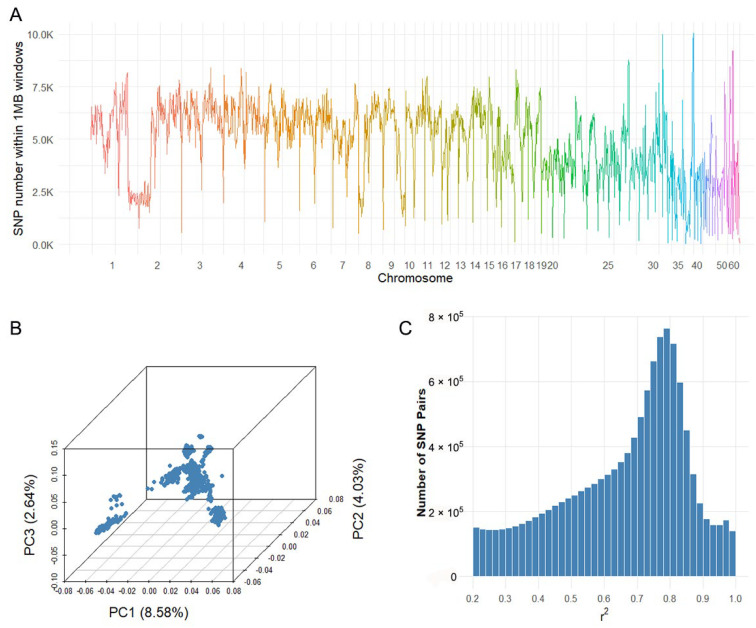
(**A**) Distribution of single nucleotide polymorphism (SNP) density. Different colors represent different chromosomes; (**B**) principal component analysis (PCA) of the first three principal components; (**C**) linkage disequilibrium (LD) r^2^ histogram (r^2^ > 0.2).

**Figure 2 ijms-26-07007-f002:**
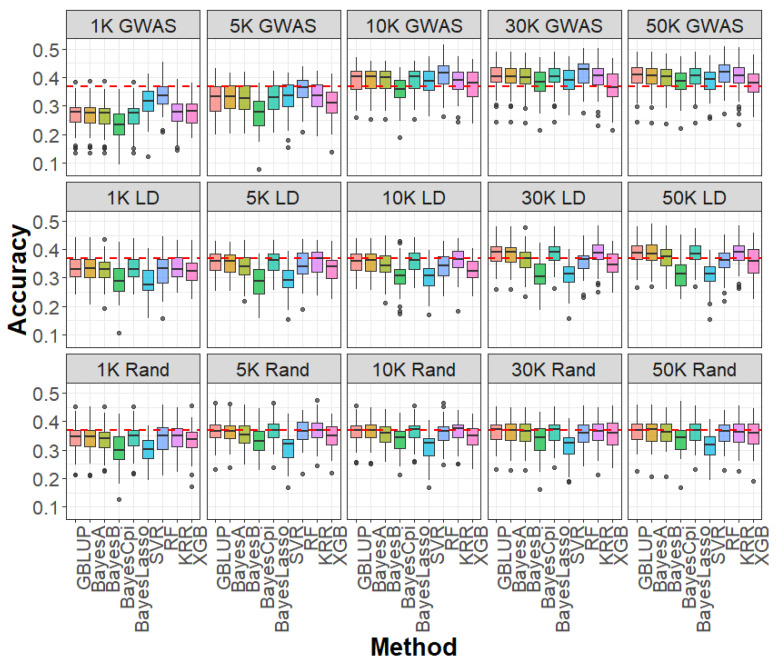
Accuracy of genomic prediction for caviar yield based on GWAS (1 K–50 K GWAS), LD pruning (1 K–50 K LD), and random sampling (1 K–50 K Rand) SNP panels, using linear models (GBLUP, BayesA, BayesB, BayesCpi, BayesLasso) and machine learning methods (SVR, RF, KRR, XGB). GWAS: genome-wide association study; LD: linkage disequilibrium; Rand: random; SVR: support vector regression; RF: random forest; KRR: kernel ridge regression; XGB: extreme gradient boosting.

**Figure 3 ijms-26-07007-f003:**
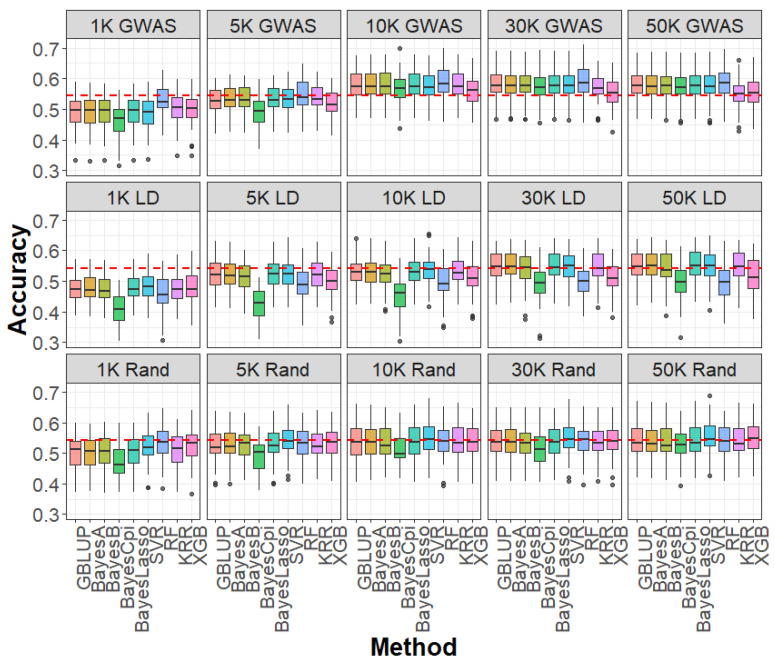
Accuracy of genomic prediction for caviar color based on GWAS (1 K–50 K GWAS), LD pruning (1 K–50 K LD), and random sampling (1 K–50 K Rand) SNP panels, using linear models (GBLUP, BayesA, BayesB, BayesCpi, BayesLasso) and machine learning methods (SVR, RF, KRR, XGB). GWAS: genome-wide association study; LD: linkage disequilibrium; Rand: random; SVR: support vector regression; RF: random forest; KRR: kernel ridge regression; XGB: extreme gradient boosting.

**Figure 4 ijms-26-07007-f004:**
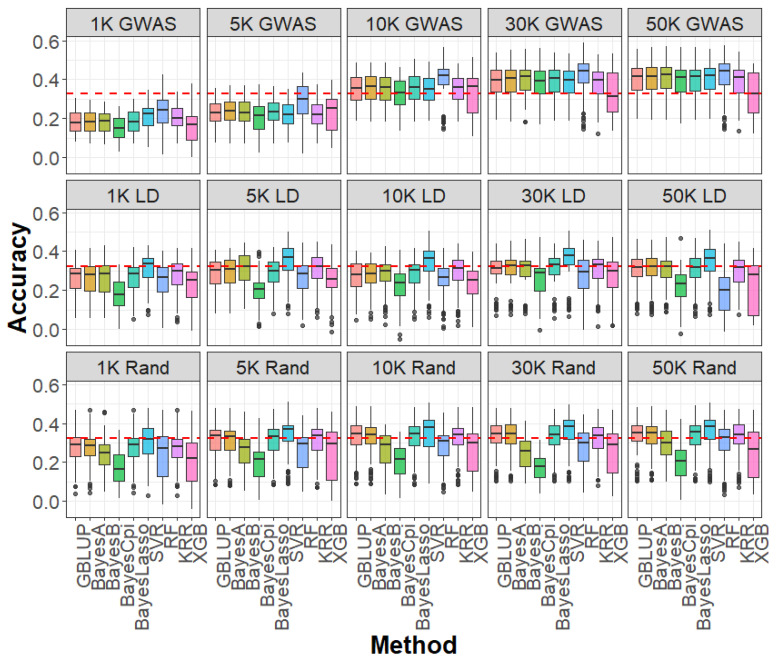
Accuracy of genomic prediction for body weight based on GWAS (1 K–50 K GWAS), LD pruning (1 K–50 K LD), and random sampling (1 K–50 K Rand) SNP panels, using linear models (GBLUP, BayesA, BayesB, BayesCpi, BayesLasso) and machine learning methods (SVR, RF, KRR, XGB). GWAS: genome-wide association study; LD: linkage disequilibrium; Rand: random; SVR: support vector regression; RF: random forest; KRR: kernel ridge regression; XGB: extreme gradient boosting.

**Figure 5 ijms-26-07007-f005:**
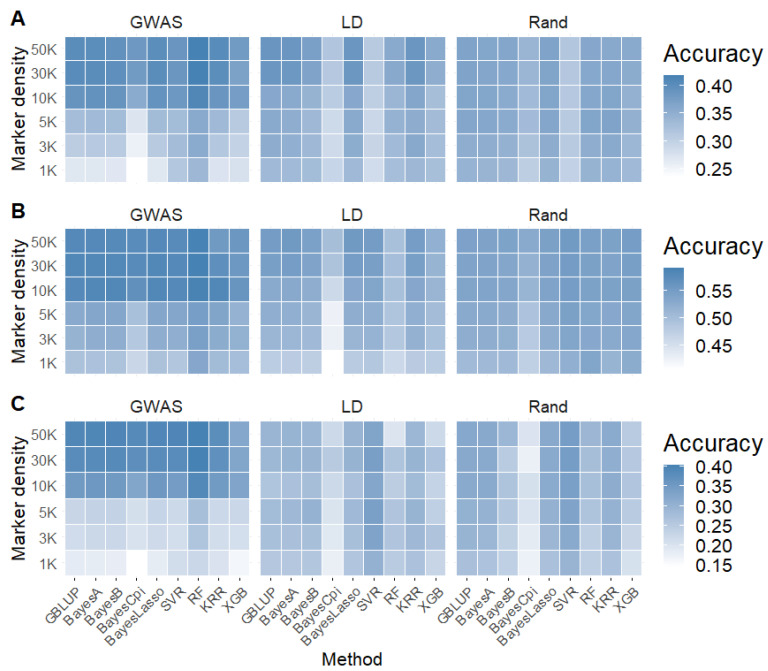
Comparison of genomic prediction accuracy for (**A**) caviar yield, (**B**) caviar color, and (**C**) body weight based on GWAS, linkage disequilibrium (LD), and random SNP selection strategies across different SNP densities, using linear and machine learning models.

**Figure 6 ijms-26-07007-f006:**
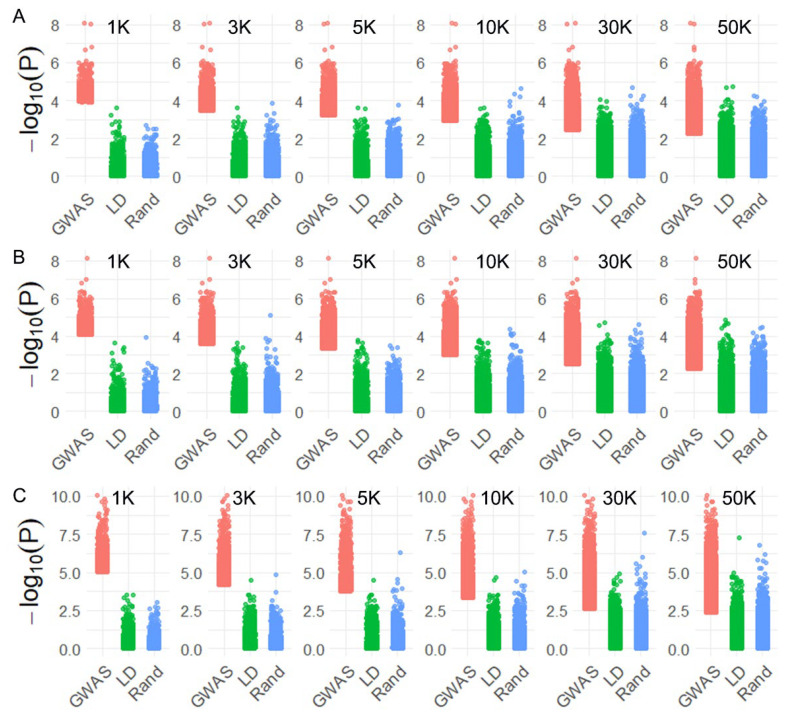
*p*-value distribution based on GWAS for (**A**) caviar yield, (**B**) caviar color, and (**C**) body weight, using GWAS, linkage disequilibrium (LD), and random SNP selection strategies across different SNP densities.

**Table 1 ijms-26-07007-t001:** Descriptive statistics of three traits in sturgeon. Caviar yield is shown as a proportion; caviar color was assessed on a 1–4 scale; body weight is reported in kilograms (kg).

Trait	N	Mean	SD	CV (%)	Max	Min
Caviar yield	971	0.193	0.057	29.414	0.439	0.021
Caviar color	971	2.398	0.642	26.789	4.000	1.000
Body weight	971	19.806	5.096	25.732	116.800	9.700

N: sample size; SD: standard deviation; CV: coefficient of variation; Max: maximum value; Min: minimum value.

**Table 2 ijms-26-07007-t002:** Percent of phenotypic variation explained (PVE, %) for caviar yield, caviar color, and body weight by SNPs selected using GWAS, LD, and random strategies across different SNP densities.

SNP Density	Caviar Yield			Caviar Color			Body Weight		
	GWAS	LD	Random	GWAS	LD	Random	GWAS	LD	Random
1 K	0.219	0.012	0.012	0.171	0.015	0.012	0.279	0.018	0.012
3 K	0.722	0.028	0.039	0.449	0.038	0.037	0.697	0.045	0.037
5 K	1.070	0.096	0.063	0.697	0.063	0.060	1.055	0.074	0.062
10 K	1.369	0.166	0.087	1.257	0.131	0.121	1.838	0.151	0.123
30 K	3.092	0.548	0.382	3.145	0.383	0.366	4.318	0.419	0.362
50 K	5.249	0.961	0.543	4.757	0.645	0.606	6.314	0.676	0.607

GWAS: genome-wide association study; LD: linkage disequilibrium.

## Data Availability

The sturgeon genotype and phenotype data are available from the corresponding author upon reasonable request for academic, non-commercial use.

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
