# Peer review of "Prioritized SNP Selection from Whole-Genome Sequencing Improves Genomic Prediction Accuracy in Sturgeons Using Linear and Machine Learning Models"

_ijms, 2025, doi:10.3390/ijms26147007_

Round 1

Reviewer 1 Report

Comments and Suggestions for Authors

MANUSCRIPT IJMS-2025-3735047 PEER REVIEW

This study explores genomic prediction in aquaculture breeding, focusing on economically important traits in Russian sturgeon—caviar yield, caviar color, and body weight. Using whole-genome sequencing data from 971 individuals, the researchers evaluated three SNP selection strategies: GWAS-based prioritization, linkage disequilibrium (LD) pruning, and random sampling. They tested various linear and machine learning models across different SNP densities to identify the most accurate prediction methods. GWAS-based SNP selection, especially at moderate densities (≥10K), consistently yielded the highest accuracy, improving prediction by up to 3.4% compared to full WGS data. LD-based selection performed comparably at higher SNP densities. Among prediction models, machine learning—particularly random forest—outperformed linear approaches, enhancing accuracy by 2.0% when paired with GWAS-based SNPs. The findings suggest that combining whole-genome sequencing, GWAS-driven SNP selection, and machine learning provides a powerful strategy to improve genomic predictions in sturgeon breeding programs.

The manuscript is well written. I detected no English language issues whatsoever. Some minor but important revisions and suggestions are provided in this review. I think there is plenty of evidence of rigor in the presentation of the results, both in terms of a scientific contribution and in the journal guidelines for authors. Specific comments are provided.

INTRODUCTION

Lines 74-75. Please elaborate on the “accumulating evidence” you are talking about. It is not clear why this is important.

Lines 75-76. Also, what this “noise” consists of, and how it potentially reduces the accuracy of genomic prediction? A brief description would be enough. The kind of explanations you made in lines 77-80 is a good example.

Line 88. Please define GWAS at first mention.

DISCUSSION

Lines 240-242. I recommend the authors include examples thereof (not just the references). A couple of examples on why “utilizing all available markers from whole-genome sequencing data does not necessarily improve, and may even reduce, the accuracy of genomic prediction”, will allow the reader to rapidly understand the reasons, based on evidence, of such a sentence.

Line 255. Please describe what FDR means at first mention.

MATERIALS AND METHODS

Lines 362-363. Please consider that, prior to 2025, the sterlet was classified under the genus Acipenser; however, this grouping was later determined to be paraphyletic, and the species is now more accurately assigned to the genus Huso. See the following references:

  • Eschmeyer, W. N.; Fricke, R. & van der Laan, R. (eds.). Species in the genus Huso. Catalog of Fishes. California Academy of Sciences.
  • Brownstein, C. D.; Near, T. J. 2025.Toward a Phylogenetic Taxonomy of Sturgeons (Acipenseriformes: Acipenseridae). Bulletin of the Peabody Museum of Natural History. 66 (1). doi:10.3374/014.066.0101. ISSN 0079-032X.

Minor and typos

Table 1. Please consider that it seems rather odd to have “statistical measures” of a “subjectively assessed scale”, such as color. Also, the weight units of measure are missing.

Fig 1a. What does the color set represent?

Data Availability Statement

Lines 529-530. I suggest the authors to better explain what “reasonable request” does mean.

Author Response

This study explores genomic prediction in aquaculture breeding, focusing on economically important traits in Russian sturgeon—caviar yield, caviar color, and body weight. Using whole-genome sequencing data from 971 individuals, the researchers evaluated three SNP selection strategies: GWAS-based prioritization, linkage disequilibrium (LD) pruning, and random sampling. They tested various linear and machine learning models across different SNP densities to identify the most accurate prediction methods. GWAS-based SNP selection, especially at moderate densities (≥10K), consistently yielded the highest accuracy, improving prediction by up to 3.4% compared to full WGS data. LD-based selection performed comparably at higher SNP densities. Among prediction models, machine learning—particularly random forest—outperformed linear approaches, enhancing accuracy by 2.0% when paired with GWAS-based SNPs. The findings suggest that combining whole-genome sequencing, GWAS-driven SNP selection, and machine learning provides a powerful strategy to improve genomic predictions in sturgeon breeding programs.

The manuscript is well written. I detected no English language issues whatsoever. Some minor but important revisions and suggestions are provided in this review. I think there is plenty of evidence of rigor in the presentation of the results, both in terms of a scientific contribution and in the journal guidelines for authors. Specific comments are provided.

INTRODUCTION

Comments 1:Lines 74-75. Please elaborate on the “accumulating evidence” you are talking about. It is not clear why this is important.

Response1:Thank you for pointing this out. We agree that the original expression “accumulating evidence” was vague. To improve clarity and precision, we have revised it. Please see L67.

Comments 2: Lines 75-76. Also, what this “noise” consists of, and how it potentially reduces the accuracy of genomic prediction? A brief description would be enough. The kind of explanations you made in lines 77-80 is a good example.

Response 2:Thank you for pointing this out. We agree with this comment. We have added a brief explanation in the revised manuscript to better illustrate this point, similar to the example in lines 77–80. Please see L69-71.

Comments 3: Line 88. Please define GWAS at first mention.

Response 3:Thank you for pointing this out. We have revised it.

DISCUSSION

Comments 4: Lines 240-242. I recommend the authors include examples thereof (not just the references). A couple of examples on why “utilizing all available markers from whole-genome sequencing data does not necessarily improve, and may even reduce, the accuracy of genomic prediction”, will allow the reader to rapidly understand the reasons, based on evidence, of such a sentence.

Response 4:Thank you for pointing this out. We agree with this comment. In the revised manuscript, we have included specific examples to support this statement. Please see L214-220.

Comments 5: Line 255. Please describe what FDR means at first mention.

Response 5: Thank you for pointing this out. We have revised it.

MATERIALS AND METHODS

Comments 6: Lines 362-363. Please consider that, prior to 2025, the sterlet was classified under the genus Acipenser; however, this grouping was later determined to be paraphyletic, and the species is now more accurately assigned to the genus Huso. See the following references:

  • Eschmeyer, W. N.; Fricke, R. & van der Laan, R. (eds.). Species in the genus Huso. Catalog of Fishes. California Academy of Sciences.
  • Brownstein, C. D.; Near, T. J. 2025.Toward a Phylogenetic Taxonomy of Sturgeons (Acipenseriformes: Acipenseridae). Bulletin of the Peabody Museum of Natural History. 66 (1). doi:10.3374/014.066.0101. ISSN 0079-032X.

Response 6: Thank you for your insightful comment and for providing relevant references. We appreciate the clarification regarding the updated phylogenetic placement of the sterlet. In light of recent taxonomic revisions (e.g., Brownstein & Near, 2025), we have updated the manuscript to reflect that the sterlet (Huso ruthenus) was formerly classified under the genus Acipenser but has now been reassigned to Huso due to the paraphyletic nature of the previous grouping. This correction has been made in the revised version to ensure taxonomic accuracy. Please see L338.

Minor and typos

Comments 7: Table 1. Please consider that it seems rather odd to have “statistical measures” of a “subjectively assessed scale”, such as color. Also, the weight units of measure are missing.

Response 7:Thank you for your insightful comment. Caviar yield is expressed as a proportion (ratio of caviar weight to body weight) and therefore has no unit. Caviar color was assessed using a 1–4 subjective scoring scale, following standard industry grading practices; as such, it also has no physical unit. Body weight is expressed in kilograms (kg), and we have now clarified this in the table caption for clarity.We have revised the table caption as follows: Table 1. Descriptive statistics of three traits in sturgeon. Caviar yield is shown as a proportion; caviar color was assessed on a 1–4 scale; body weight is reported in kilograms (kg). Please see L98-99.

Comments 8: Fig 1a. What does the color set represent?

Response 8:Thank you for pointing this out. Different colors represent different chromosomes; We have revised it. Please see L117-118.

.

Data Availability Statement

Comments 9: Lines 529-530. I suggest the authors to better explain what “reasonable request” does mean.

Response 9:Thank you for your valuable suggestion. In response, we have revised the Data Availability Statement to clarify the meaning of “reasonable request.” Specifically, we now state that the sturgeon genotype and phenotype data are available from the corresponding author upon reasonable request for academic, non-commercial use. Please see L496-497.

Reviewer 2 Report

Comments and Suggestions for Authors

All abbreviations should be tabulated( huge number )

Conclusion more  than enough—

Materials and methods should be written before the results and after introduction—also it is very long and not well-organized

Very long topic ?

What about the simple summary as all MDPI journals ?

Abstract :

There are no highlights—why

There is no graphical abstract

What is /are the creativity of this work ??

LN/15-16---SNPs—more details are requested (its role )

LN/19—why did you select these 3 traits ?

LN/20---clarify this in detail

LN/21---reduced marker density—explain how ?

There is no conclusion

Divide the abstract into backgrounds/aims/methods/results and discussion

LN/32—add aquaculture industry/breeding/caviar traits to the keywords

Introduction

LN/36/39/41/42/46/49—etc.----add references

LN/37—explain this in detailed manner

LN/40---add value not profile

The introduction is extremely very long and repeated—rewrite it again

Aims should be more clarified

Novelty needs to be more highlighted

Materials and methods

LN/342—very old date

LN/344—standardized conditions—mention all

There is no plan for the study area

The most descriptive methodologies are without references

What about the statistical analysis

M&M is very long and not -well-organized —rewrite it again

There is no ethical approval code—should be

Results

LN/101---moderate----------individuals----why ?

LN/104---which type (s) of pigments do you mean

LN/111—what about the size/weight/distribution of the fish –so M&M should be before the results

LN/118—clustered closely—how/pathogenesis ?

Why did you not do histological sections or chromosomal studies for more confirmation

Are there anu gross abnormalities were detected

What about the clinical signs of the investigated fish

Are there any detected mortalities and why ?

LN/134---method---what is this

Figures need to be more described and cleared

Results are very long

Discussion

It is very long

You should discuss your results with the previous investigators

Rewrite it again

Conclusion

More  than enough

References

Some cited references need to be more update

Huge number of references were used(39)why ???

Some cited references with missing data

Some cited references contained more than 6 authors—why ? should be 6 at the maximum plus et al. with the last ones –apply for all (ref—6/7/8/10/19/21—etc. )

Some journal names were written abbreviated , while others were not—why ? same style should be –apply for all

All references should be rewritten

There are no gross figures

Comments on the Quality of English Language

Okay

Author Response

Comments 1: All abbreviations should be tabulated( huge number )

Response 1:Thank you for your suggestion. We have carefully checked the manuscript and added definitions for all abbreviations at their first appearance to enhance readability. Please see L499-520.

Comments 2: Conclusion more  than enough—

Response 2: Thank you for your comment. We have revised the Conclusion section to make it more concise and improve readability. Please see L479-485.

Comments 3: Materials and methods should be written before the results and after introduction—also it is very long and not well-organized

Response 3: Thank you for the suggestion. According to the journal’s format requirements, the Materials and Methods section is placed after the Results section. Nevertheless, we have carefully revised and reorganized the Materials and Methods section to improve clarity and readability.

Comments 4: Very long topic ?

Response 4: Thank you for your comment. We acknowledge that some sections may be lengthy, and we have carefully reviewed the manuscript to improve conciseness and clarity throughout the text.

Comments 5: What about the simple summary as all MDPI journals ?

Response 5: Thank you for your suggestion. However, the IJMS journal does not require a Simple Summary for submission. Therefore, we have not included it in our manuscript.

Abstract :

Comments 6:

There are no highlights—why

There is no graphical abstract

Response 6: Thank you for your valuable comments. We appreciate the suggestion regarding highlights and a graphical abstract. However, the journal IJMS does not require authors to provide highlights or a graphical abstract for submission. Therefore, these elements were not included in our manuscript.

Comments 7:

What is /are the creativity of this work ??

LN/15-16---SNPs—more details are requested (its role )

LN/19—why did you select these 3 traits ?

LN/20---clarify this in detail

LN/21---reduced marker density—explain how ?

There is no conclusion

Divide the abstract into backgrounds/aims/methods/results and discussion

LN/32—add aquaculture industry/breeding/caviar traits to the keywords

Response 7: Thank you very much for your detailed and constructive comments. We have carefully revised the abstract to address all the points you raised. Specifically, we have provided more details on SNPs and their role, clarified the rationale for selecting the three traits, and explained how marker density was reduced. The abstract is now structured into clear sections: background, aims, methods, results, and discussion, to enhance clarity and completeness. Additionally, we have added keywords related to the aquaculture industry, breeding, and caviar traits. We believe these revisions have improved the clarity and organization of the manuscript and better highlight the novelty and practical significance of our work. We sincerely appreciate your valuable suggestions. Please see the newly submitted version.

Introduction

Comments 8: LN/36/39/41/42/46/49—etc.----add references

Response 8: Thank you for your suggestion. We have added the relevant references.

Comments 9: LN/37—explain this in detailed manner

Response 9: Thank you for your comment. We have revised it to provide a more detailed explanation for better clarity. Please see L42-45.

Comments 10: LN/40---add value not profile

Response 10: Thank you for your comment. Done.

Comments 11: The introduction is extremely very long and repeated—rewrite it again

Response 11: Thank you for your valuable feedback. We have carefully revised and significantly shortened the introduction to eliminate repetitions and improve clarity and focus. Please see the newly submitted version.

Comments 12:

Aims should be more clarified

Novelty needs to be more highlighted

Response 12: Thank you for your insightful comments. We have clarified the aims of the study more explicitly and highlighted the novelty in the revised manuscript to better emphasize the unique contributions of our work. Please see L80-87.

Materials and methods

Comments 13: LN/342—very old date

Response 13: Thank you for your comment. Due to the long sexual maturity period of sturgeon, which is approximately 8 years, the full-sibling families were established in 2012, and the F1 generation was obtained in 2020. We have added this explanation in the revised manuscript for clarity. Please see L318-320.

Comments 14: LN/344—standardized conditions—mention all

Response 14: Thank you for your suggestion. We have clarified the standardized conditions in the revised manuscript. Please see L320-323.

Comments 15: There is no plan for the study area

Response 15: We thank the reviewer for pointing out this omission. In the revised manuscript (Section 4.1), we have clarified the rationale for selecting the study site. Specifically, we emphasized that the fish were sampled from a large-scale commercial sturgeon breeding base in Zhejiang, China, which maintains standardized aquaculture practices and long-term pedigree records. This facility represents a typical sturgeon farming environment in eastern China and provides reliable conditions for genomic evaluation. These additions now better reflect the representativeness and planning of the study area. Please see L316-318.

Comments 16: The most descriptive methodologies are without references

Response 16: Thank you for the valuable comment. We have added the relevant references to the descriptive methodologies in the revised manuscript. Please see L369, L419, L437 and L449-451.

Comments 17: What about the statistical analysis

Response 17: We thank the reviewer for this valuable comment. In the revised manuscript (Section 4.7), we have clarified the statistical analysis procedures used to evaluate the performance of genomic prediction models. Specifically, we implemented a repeated fivefold cross-validation strategy to assess model robustness, and used multiple statistical metrics—including Pearson correlation, regression coefficient, mean squared error (MSE), and mean absolute error (MAE)—to comprehensively evaluate prediction accuracy, bias, and precision. These methods allow for a robust comparison of prediction models and marker selection strategies across different SNP densities.

Comments 18: M&M is very long and not -well-organized —rewrite it again

Response 18: Thank you for your comment. We have thoroughly revised and reorganized the Materials and Methods section to improve clarity and conciseness. Please see the newly submitted version.

Comments 19: There is no ethical approval code—should be

Response 19: Thank you for pointing this out. Our study only involves the use of fin samples for DNA extraction and sequencing, and does not involve any procedures requiring animal ethical approval. Therefore, we believe that ethical approval is "Not applicable" to this study. Please see L323 and L337.

Results

Comments 20: LN/101---moderate----------individuals----why ?

Response 20: Thank you for your comment. We have removed the subjective term “moderate” and instead reported variability using coefficient of variation (CV) values, which offer a quantitative and comparable measure of variation across traits.

Comments 21: LN/104---which type (s) of pigments do you mean

Response 21: We appreciate the question. In this study, caviar color was assessed using a standardized 4-point visual scoring system, ranging from dark to gold. All scores were assigned by a single trained operator using a reference image, ensuring consistency. This approach reflects overall pigment appearance but does not quantify specific pigment types. Please see L329-331.

Comments 22: LN/111—what about the size/weight/distribution of the fish –so M&M should be before the results

Response 22: Thank you for your insightful suggestion. All 971 fish used in this study were of the same age to minimize age-related variation. Body weight data, including range and variability, are presented in Table 1. As required by the journal’s formatting guidelines, the Results and Discussion section is placed before the Materials and Methods section.

Comments 23: LN/118—clustered closely—how/pathogenesis ?

Response 23: Thank you for your observation. The term “clustered closely” refers to the distribution pattern of specific phenotypic traits (e.g., caviar color), which showed limited variation within certain ranges. This clustering is attributed to shared genetic background and uniform rearing conditions rather than any pathological process.

Comments 24: Why did you not do histological sections or chromosomal studies for more confirmation

Response 24: We appreciate your suggestion. Histological or cytogenetic analyses were not included in the current study, as the main focus was on genome-wide SNP-based GWAS and GS analysis for economically relevant traits in a healthy aquaculture population. All individuals were reared under standard farming conditions, with no indication of reproductive or developmental abnormalities. However, we agree that future work integrating histological and cytogenetic data could provide additional insights, especially into trait biology.

Comments 25: Are there anu gross abnormalities were detected

Response 25: Thank you for pointing this out. No gross abnormalities were observed among the investigated fish. All individuals appeared phenotypically normal and were routinely inspected during rearing for signs of external deformities or disease. Only healthy individuals were included in this study.

Comments 26: What about the clinical signs of the investigated fish

Response 26: Thank you for pointing this out. No clinical signs of disease were observed in any of the fish sampled. Regular health monitoring was performed during routine aquaculture practices, and all individuals included in this study were considered clinically healthy at the time of sampling.

Comments 27: Are there any detected mortalities and why ?

Response 27: Thank you for pointing this out. No abnormal mortality events occurred during the sampling period. The fish were maintained under standard aquaculture conditions, and routine health surveillance indicated good overall population health. Minor mortality unrelated to the study occasionally occurred as part of normal farm operation but was not linked to any pathological condition.

Comments 28: LN/134---method---what is this

Response 28: Thank you for your valuable comment. The figure in this section illustrates the genomic prediction method used in this study. We have revised and clarified the figure legend to explicitly describe the methodology and its components, ensuring the approach is understandable. Please see L131-136, L138-143, L145-150, we hope this resolves any confusion.

Comments 29: Figures need to be more described and cleared

Response 29: Thank you for pointing this out. We have revised it to make it clearer, please see L131-136, L138-143, L145-150.

Comments 30: Results are very long

Response 30: Thank you for your valuable comment. We have carefully revised the Results section by streamlining the content and improving its organization to enhance readability and focus. Please see the newly submitted version.

Comments 31:

Discussion

It is very long

You should discuss your results with the previous investigators

Rewrite it again

Response 31: Thank you for your constructive comment. We appreciate the reviewer’s suggestions regarding the Discussion section. We have carefully revised the Discussion section by shortening the content, improving its structure, and integrating comparisons with relevant previous studies to better contextualize our findings. The updated version now provides a more concise, focused, and evidence-based discussion, highlighting the significance and implications of our results.

Comments 32:

Conclusion

More  than enough

Response 32: Thank you for your comment. We have revised the Conclusion section to make it more concise and improve readability. Please see L479-485.

Comments 33:

References

Some cited references need to be more update

Huge number of references were used(39)why ???

Some cited references with missing data

Some cited references contained more than 6 authors—why ? should be 6 at the maximum plus et al. with the last ones –apply for all (ref—6/7/8/10/19/21—etc. )

Some journal names were written abbreviated , while others were not—why ? same style should be –apply for all

All references should be rewritten

Response 33: Thank you for your valuable comments. We have carefully revised and standardized the reference section to fully comply with the journal's formatting requirements. Missing data have been completed, journal names have been consistently formatted. Regarding the total number of references cited in our manuscript, we aimed to provide comprehensive support for our methodology and discussion. Similar citation volumes are common in this journal—for example, Wang et al. (2025) cited 71 references, Zhu et al. (2025) cited 68, and Zhou et al. (2025) cited 42. We hope the revised version meets your expectations, and we sincerely appreciate your time and helpful feedback.

Comments 34: There are no gross figures

Response 34: Thank you for your comment. While we acknowledge that comprehensive figures can be useful, our manuscript presents detailed comparative results through multiple tables and figures in the main text and supplementary materials

Reviewer 3 Report

Comments and Suggestions for Authors

The manuscript is a well written well conceived peice of work. but this needs some modifications

Line 110: As the sturgeon fish has a genome data submitted in NCBI why the author didnot took that as the reference?

Figure 2,3,4: the legend font should be decreased in size to accomodate it to its corresponding points. Also all the legends must explained properly.

Line 219: Please check the sentence grammar and sentence formation.

The authors should clearly describe the each parameter they used and their usefulness as the layman cannot understand the difference between GWAS-selected SNPs and BayesC and others.

Author Response

The manuscript is a well written well conceived peice of work. but this needs some modifications

Comments 1: Line 110: As the sturgeon fish has a genome data submitted in NCBI why the author didnot took that as the reference?

Response 1: Thank you for your comment. As correctly noted, the sterlet reference genome (assembly ASM1064508v1) used in our study is indeed publicly available in the NCBI database, and we have utilized this genome as our reference (Reference [30]). Due to the segmental rediploidization characteristic of sturgeon genomes and the lack of a high-quality polyploid reference genome for Russian sturgeon, the diploid sterlet genome serves as an appropriate and commonly used proxy for genomic analyses in polyploid sturgeon species. We have clarified this in the revised manuscript to avoid confusion. Please see L338-342.

Comments 2: Figure 2,3,4: the legend font should be decreased in size to accomodate it to its corresponding points. Also all the legends must explained properly.

Response 2: Thank you for your suggestion. We used a relatively larger legend font size to ensure clarity and visibility. However, we have carefully revised the figure legends to make all information and symbols clearly explained. Please refer to the updated figure legends in the revised manuscript at L131–136, 138–143, and 145–150.

Comments 3: Line 219: Please check the sentence grammar and sentence formation.

Response 3: Thank you for your comment. We have revised the sentence to improve its grammar and clarity. Please see the updated version in the revised manuscript at L205–207.

Comments 4: The authors should clearly describe the each parameter they used and their usefulness as the layman cannot understand the difference between GWAS-selected SNPs and BayesC and others.

Response 4: Thank you for your valuable suggestion. We have revised the manuscript to clearly describe each parameter used in the analysis and explained their relevance and differences, including the distinction between GWAS-selected SNPs, BayesC, and other methods. These revisions aim to improve clarity and make the content more accessible to a broader readership. Please see L396-402, L413-417, L449-456.

Round 2

Reviewer 2 Report

Comments and Suggestions for Authors

Accepted